# A Simple Practical Guide to Genomic Diagnostics in a Pediatric Setting

**DOI:** 10.3390/genes12060818

**Published:** 2021-05-27

**Authors:** Alan Taylor, Zeinab Alloub, Ahmad Abou Tayoun

**Affiliations:** 1Al Jalila Genomics Center, Al Jalila Children’s Specialty Hospital, Dubai, United Arab Emirates; Ahmad.Tayoun@ajch.ae; 2Neurodevelopment Department, Al Jalila Children’s Specialty Hospital, Dubai, United Arab Emirates; ZAlloub@ajch.ae; 3Center for Genomic Discovery, Mohammed Bin Rashid University of Medicine and Health Sciences, Dubai, United Arab Emirates

**Keywords:** genetic testing, pediatrics, diagnostics, next generation sequencing, microarray, whole exome sequencing, whole genome sequencing, precision medicine

## Abstract

With limited access to trained clinical geneticists and/or genetic counselors in the majority of healthcare systems globally, and the expanding use of genetic testing in all specialties of medicine, many healthcare providers do not receive the relevant support to order the most appropriate genetic test for their patients. Therefore, it is essential to educate all healthcare providers about the basic concepts of genetic testing and how to properly utilize this testing for each patient. Here, we review the various genetic testing strategies and their utilization based on different clinical scenarios, and test characteristics, such as the types of genetic variation identified by each test, turnaround time, and diagnostic yield for different clinical indications. Additional considerations such as test cost, insurance reimbursement, and interpretation of variants of uncertain significance are also discussed. The goal of this review is to aid healthcare providers in utilizing the most appropriate, fastest, and most cost-effective genetic test for their patients, thereby increasing the likelihood of a timely diagnosis and reducing the financial burden on the healthcare system by eliminating unnecessary and redundant testing.

## 1. Introduction

The decreasing sequencing cost and the rapid expansion of our knowledge of human disease genes has fueled a significant increase in the uptake and utilization of clinical genomic testing, to the extent that nearly all healthcare providers are required to order genetic testing as part of their clinical practice. Traditionally, all patients requiring genetic testing would be referred to the genetics department, where they would be evaluated and the most appropriate test would be ordered. Such practice was partly stymied by the available technology, which only allowed iterative sequential targeted testing approaches, leading to costly and time-consuming diagnostic odysseys. However, with the advances in technology, the use of genetic testing within the clinic has since far outpaced the number of clinical geneticists and genetic counselors required to see all patients who will need this testing within a reasonable timeframe. One study in the United States, which has the second highest number of genetic counselors per capita (second only to Cuba), predicted a deficit of genetic counselors in the coming years [1]. This finding indicates that there will be a global shortage of trained genetic counselors to meet the growing demands of genetic testing across all aspects of healthcare for many years, as approximately 4500 genetic counselors, which represents ~60% of the entire global profession, are based in North America [2]. The remaining ~40% of practicing genetic counselors are distributed unevenly across the globe, with certain regions having a higher density of genetic counselors per capita, most notably the United Kingdom, with approximately 300 genetic counselors, 450 in the rest of Europe, 220 in Australia and New Zealand, approximately 230 in Japan, 120 in Taiwan, and 80 in Israel. However, many regions, such as China and most countries in South America, do not yet recognize genetic counseling as an independent occupation and counseling is provided by other healthcare workers [2].

Genetic testing was traditionally preceded by a thorough physical examination, family history assessment, and possibly other diagnostic workup by a clinical geneticist who would generate a differential diagnosis list based on all findings. The suspected conditions would then be tested “one by one” or at least “gene by gene”, an approach, which besides being costly and time consuming, is often ineffective due to the genetic heterogeneity of most inherited conditions. In the advent of affordable next generation sequencing (NGS), the ability of simultaneously sequencing a large number of genes (so-called gene panels) up to whole exome sequencing (WES) introduced a major shift from the classical sequential testing strategy into fast and more cost-effective ‘genomic’ testing paradigms, oftentimes putting an end to lengthy diagnostic odysseys.

The utility of genomic testing goes beyond finding a diagnosis. This testing has always enabled healthcare providers to accurately determine diagnosis, prognosis, and recurrence risks and, in the past decade, has been powerful in identifying patients who are candidates for the rapidly growing personalized targeted treatments and gene therapy. As such, this expansion of effective treatments for previously uncurable genetic conditions has increased the utility of genetic testing. This fact, combined with the limited access to clinical geneticists, suggest that most, if not all, healthcare providers must become familiar with, or obtain the necessary support for the genetic testing process.

In this review, we provide an introduction and overview of the current genomic testing landscape and possible testing strategies, which take into consideration clinical presentations and several tests’ performance characteristics, costs, and limitations.

## 2. Genomic Testing Strategies

Findings from the patient’s clinical assessment, including physical exam, family history, and/or other laboratory or imaging findings, will determine the best genomic testing strategy, which takes into consideration technology (sequencing, microarrays, methylation, etc.), diagnostic yield, cost, turnaround time, and whether testing should be targeted to a few genes or scaling the analysis to include the exome of genome in search of diagnosis (Figure 1 and Table 1).

For conditions with very clearly defined, distinct features, targeted testing, using the appropriate technology, would be pursued. For example, Achondroplasia can typically be clinically diagnosed in the newborn period with the findings of proximal shortening of the arms, large head, narrow chest, and short fingers. For this condition, targeted analysis of a single sequence variant in the *FGFR3* gene would be the fastest, most cost-effective, and most sensitive testing strategy to be pursued. On the other hand, for a patient who is highly suspected to have Osteogenesis Imperfecta (OI), a disease where large number of sequence variants in the *COL1A1* and *COL1A2* gene can be causative, a sequencing panel targeting those two genes would be most appropriate. A male boy with intellectual disability and a maternal family history of premature ovarian failure might be best suited for fragile X testing which is designed to enumerate the ‘CGG’ triplet number in the promoter region of the *FMRP* gene. Finally, a patient with features consistent with DiGeorge syndrome will require targeted deletion analysis of the genomic region encompassing the long arm of chromosome 22 (22q11.2). It is important to note that in the last two examples, sequencing methodologies would most likely have failed to identify the fragile X repeat expansion or the DiGeorge 22q11.2 deletion, highlighting the importance of selecting the appropriate technology even when pursuing targeted testing.

For patients with broader, non-specific clinical presentations, other comprehensive genomic testing approaches will be more appropriate, as discussed below.

### 2.1. Chromosomal Microarray

Chromosomal microarray (CMA) is a broad, nonspecific, genomic test which analyzes the human chromosomes for large deletions and duplications (also called copy number variants or CNVs) of DNA. For over a decade, CMA has been considered the first-tier genetic test for children with multiple congenital anomalies (MCA), intellectual disability (ID) or developmental delay (DD), and autism spectrum disorders (ASD). CMA has replaced the karyotype for children with these indications, largely due to the higher resolution and, therefore, higher diagnostic yield (15–20%) for a microarray compared to a karyotype (~3%) [3] (Table 1). Typically, a karyotype can detect a copy number variant that is 3–5 Mb in size or larger, while a microarray would detect deletions down to 25 kb—and possibly higher depending on probe coverage for specific genomic regions. In addition to performing full CMA analysis for broad indications such as ID, DD, MCA, and ASD, targeted deletion testing for conditions like DiGeorge (AKA 22q11.2 deletion syndrome) can be performed by focusing the analysis on the relevant region—22q11.2 for the DiGeorge example—while masking all other genomic regions. If the clinical team is confident of a specific microdeletion or microduplication condition, then ordering a targeted microarray can be a cheaper option with a faster turnaround time, while enabling a reflex to full chromosomal analysis if the targeted test was negative. It must be noted that not all genetic testing laboratories will offer targeted CMA analysis and may only offer a full CMA as standard, so it will be important to understand the laboratory’s offering before ordering a CMA targeted test.

While CMA platforms utilize sophisticated equipment to analyze chromosomes for CNVs via molecular probes designed to be specific to the human genome, a karyotype is a visual analysis of the chromosomes under a microscope. Therefore, the resolution of a karyotype is significantly lower, since only large CNVs can be visually detected. However, a karyotype has one advantage over CMA, as it can identify balanced structural rearrangements such as balanced translocations and inversions, which cannot be detected by CMA. A karyotype would still be recommended as the first-tier test for certain clinical conditions or settings. For example, approximately 95% of cases of Down syndrome—caused by the addition of an extra copy of chromosome 21—are due to spontaneous meiotic nondisjunction resulting in an extra free chromosome 21 (or trisomy 21) [18].

However, in a subset of cases, Down syndrome is a result of the extra chromosome 21 being attached to chromosome 14—this is called a Robertsonian translocation (A translocation is when any partial or complete chromosome is attached to another chromosome). A CMA will only identify the extra material—chromosome 21 in the case of a Robertsonian translocation—but not the location of this extra material, whereas a karyotype can distinguish between an extra free chromosome 21 or a 21:14 translocation or any other type of translocation. This information is important to help determine recurrence risks for the family. If it was due to free trisomy 21, the recurrence risk to future pregnancies is approximately 1%, while the recurrence risk for a 21:14 translocation will be 10–15% if the mother carries the balanced translocation and will be 2–3% if the father carriers the translocation [19,20].

### 2.2. Whole Exome Sequencing

Another broad, nonspecific test that is widely utilized clinically is whole exome sequencing (WES). Whereas a CMA analyzes the genome for large CNVs, WES focuses mainly on identifying small sequence variants in coding regions of the genes. Historically, WES could only identify single nucleotide variants (SNVs) and small (<50 bp) deletions or insertions (indels); however, more recently, many laboratories developed new algorithms and computational tools to detect large deletions and duplications from NGS data [21,22]. Given its higher diagnostic yield (36%) relative to CMA (15–20%), it has been suggested that WES should be considered the first-tier test for patients with neurodevelopmental disorders [8]. If copy number calling is available for those patients through WES, this might remove the need to reflex to CMA after a negative result. Starting with a WES assay that can identify CNVs, will allow for an overall faster, more cost-effective, and more sensitive testing paradigm [8]. The diagnostic yield of WES, varies greatly depending on the clinical indication, ranging from 8% for autism spectrum disorders, 70% for epileptic encephalopathies, and 76% for primary ciliary dyskinesia [9].

Generally, WES should be considered when there is no clear diagnosis and the patient’s clinical features are not suggestive of a specific syndrome or a primary presentation which can be caused by variants in a limited number of genes for which a test (or a gene panel) is available. Additionally, a clinician should consider ordering a WES if the primary phenotype can be caused by too many genes to be reasonably analyzed as part of a panel. For example, for conditions such as nonspecific developmental delays where over 200 associated genes can be involved, WES should be the test of choice.

When ordering WES, it is strongly recommended that it is sent as part of a trio, i.e., parental samples are sent along with the patient’s sample. Parental DNA will not be directly analyzed in trio WES but will be used to understand variants’ inheritance patterns in the patient for more accurate interpretation. Specifically, the addition of the parents’ samples in the analysis will allow for the identification de novo variants, and compound heterozygous variants (inheritance of two different heterozygous variants, one from each parent, in a single gene) in the patient. According to the American College of Medical Genetics and Genomics (ACMG) and the Association for Molecular Pathology (AMP) sequence variant classification guidelines [23], if the variant is confirmed to be de novo and if paternity and maternity are also confirmed, this information is considered to be strong evidence that the variant is pathogenic. The latter can be confirmed using SNP analysis of the trio exome data, which is another benefit of including parental samples in this test. For recessive conditions, if the variants are found to be in the compound heterozygous state (i.e., two identified variants in the proband and each variant is inherited from a different parent), this information can be significant and can help classify the variants in certain circumstances. Trio analysis will also highlight variants that are associated with dominant conditions, and if those are found in the unaffected parents, they can be readily classified as benign, especially in highly penetrant and severe conditions.

Informed consent is an essential component when pursuing any type of genetic testing, but it is especially important when pursuing WES due to its complexity. The pre-test counseling session should not be considered a formality and time needs to be taken to address the families’ questions and concerns before obtaining consent.

Families need to be aware, when parental samples are collected for analysis, that there will be no report issued for the parents; only a single report will be issued for the proband. This information should be clearly communicated with the family during the pre-test counseling session.

In addition, families need to be adequately counseled about the possibility of secondary and incidental findings, both of which are unrelated to the patient’s primary indication but can be identified through WES and can have implications for the proband and/or the family.

Secondary findings comprise 59 genes which have been selected by the American College of Medical Genetics, as they are associated with conditions that are both life-threatening and also medically actionable, i.e., a medical intervention can improve outcomes [24]. The associated conditions with the 59 genes include hereditary cancer, cardiac conditions, connective tissue disorders, familial hypercholesteremia and malignant hyperthermia susceptibility. For secondary findings, the default is ‘opt-in’; hence, this information should be reviewed with the family to allow them to ‘opt-out’ if they choose to do so. It should be noted that if a secondary finding is identified in the patient, it will also be identified in one of the parents, unless it is de novo. Conversely, if the patient is negative for secondary findings, it does not necessarily mean the parents are negative, as they may carry the pathogenic variant on the other allele that was not inherited in the patient.

Incidental findings are also unrelated to the primary indication of the WES but can lead to serious health conditions for which useful medical intervention might or might not be available. Incidental findings can also include unexpected biological relationships, such as non-paternity and unlawful marriages through incest. Both such scenarios have legal, ethical, and cultural consequences which the families have to be aware of during the consent process.

It should be noted that due to the complexity of WES, the intricacies of the consenting process, and possibility of complex result interpretation, the input of a genetic specialist should be sought if possible.

### 2.3. Targeted Gene Panels

A targeted gene panel is often a more appropriate—sensitive, fast and cost-effective—first-tier test for genetically heterogeneous conditions (<100 genes) with a highly specific phenotype (e.g., Noonan syndrome, Usher syndrome, Waardenburg etc.). Gene panels are a curated, fixed lists of genes which are known to cause a specific condition or phenotype. While nearly every laboratory will include the most common genes that are associated with a given condition, and are expected to identify the highest proportion of positive cases, these lists will typically vary between each lab. This variation is based on the gene-disease evidence required by each laboratory before including the gene on a panel. Until recently, gene validity assessment was somewhat subjective and relied on information from internal databases, case reports, animal models, and other functional studies [25,26,27].

It is, therefore, important to review the genes on a panel before ordering to ensure the desired genes are included for the highest possible clinical sensitivity. Besides their relatively low cost and fast turnaround time, gene panels analysis is highly focused on the indication-relevant genes and often identify few variants for interpretation leading to a significantly lower variants of uncertain clinical significance (VUS) to report relative to WES. However, a major disadvantage of targeted gene panels is the fact that their gene content is static in nature. Any newly discovered genes would require that the panel be redesigned to include new probes targeting the novel genes (assuming they meet the clinical validity criteria to be included on the panel). This new design with then have to be re-validated before clinical implementation [28].

### 2.4. Customizable Gene Panels

To overcome the static nature of targeted gene panels while still benefiting from low cost, fast turnaround time, and low burden of interpretation and VUSs offered by targeted gene analysis, many labs now offer ‘customizable’ or ‘virtual’ gene panels in which WES is performed and then the desired indication-relevant genes are virtually—using bioinformatic tools—targeted for analysis. This approach enables the addition of novel genes as soon as they become clinically valid without the need for redesign and revalidation. Furthermore, this testing strategy has the additional flexibility to reflex to a different gene set or to the whole exome, if the original analysis was negative, without the need for a new blood sample or any additional sequencing.

Finally, this ‘customizable’ gene panel approach obviates the additional burden of secondary or incidental findings, since they are bioinformatically masked by default. It should be noted that if the gene list targeted for analysis becomes very large (greater than 200 genes), pursuing trio WES might be more appropriate. For example, isolated autism spectrum disorder, intellectual disability, or arthrogryposis have 200+ associated genes; however, including additional clinical findings can reduce the number of genes necessary for analysis. The larger the number of genes included for analysis, the higher the number of VUSs that might be reported. In this case, the benefits of a trio WES segregation analysis surpass those of the customizable gene panel, as it will empower the interpretation of VUSs using the parental samples, as discussed above.

For all the above sequencing tests, it is important to note that copy number variants (CNVs), which can significantly contribute to the diagnostic yield in certain disease areas, might not be easily detectable from NGS [29,30]. Nonetheless, most laboratories currently attempt at calling CNVs from NGS data, while applying orthogonal confirmatory methods before reporting [31,32]. It is, therefore, important to understand whether CNVs are major contributors to the pathogenic spectrum of the patient disease, and whether those CNVs can be identified before ordering the test.

### 2.5. Single Gene Analysis

Single gene analysis has become less utilized due to genetic heterogeneity for most conditions and the widespread use of NGS, which has allowed for the simultaneous analysis of multiple genes via gene panels. Gene panels have a higher yield and lower cost than a traditional single gene testing approach. Nonetheless, certain conditions with highly specific presentations and no allelic or gene heterogeneity might best be tested by targeted variant analysis or full gene sequencing. For example, 99% of achondroplasia cases are due to a single pathogenic variant (Gly380Arg) in the *FGFR3* gene, which can best be detected by a targeted variant analysis such as Sanger sequencing or PCR-based genotyping [33]. On the other hand, over 1000 sequence variants in the *CFTR* gene can cause cystic fibrosis [34]. In this case, the best testing strategy would be full gene sequencing by Sanger, or more effectively by NGS. Although a number of mutations account for the majority of cases in the Caucasian and Ashkenazi Jewish populations, full *CFTR* gene sequencing by NGS has higher sensitivity to detect all possible SNVs, and possibly CNVs, in cystic fibrosis patients with diverse ancestries, without compromising cost and turnaround time [31,35]. When in doubt of specific diagnosis, it might be a better option to order a gene panel to include other conditions on the differential, as the difference in cost and turnaround time is typically minimal compared to the increase in diagnostic yield.

There are other types of genetic changes that cannot be detected by the technologies that have been reviewed so far. CMA is used to detect large deletions and duplications, while NGS, either in the form of single gene analysis, gene panels, or whole exome sequencing, is used to identify sequence variants, and more recently intragenic CNVs. Other types of genetic variation, outlined below, underlying several disorders require different targeted testing.

### 2.6. Triplet Repeat Disorders

These disorders are caused by an expansion of triplet DNA repeats, for example CGG repeat expansion (>200 repeats) in the promoter region of *FMRP* in fragile X patients. The genes associated with these conditions typically have a normal number of triplet repeat bases [11]. However, these repeats can increase in number from one generation to the next, resulting in disease. Triplet repeat disorders are not caused by a sudden expansion of repeats from a normal range to a pathogenic one. Rather, there is often a “grey zone” between the normal variation and disease range. This ‘grey zone’ repeat number may not only have some sort of clinical significance for individuals carrying it (e.g., premutation 55–200 carriers in the *FMRP* gene), but it is also an indicator that it may be expanded in one generation to a larger number of repeats, resulting in disease. The concept of an expansion between generations is called anticipation. Examples of triplet repeat disorders include Fragile X syndrome, myotonic dystrophy, or spinocerebellar ataxia. Triplet repeats cannot be confidently identified using NGS and require specific targeted testing to be able to accurately amplify and size those repeats.

### 2.7. Methylation Disorders

Methylation (addition of -CH3 groups) is a normal epigenetic modification of the DNA rather than a direct change to the DNA sequence itself. Methylation is important for the normal regulation of transcription, embryonic development, genomic imprinting, genome stability, and chromatin structure [36]. The methylation profile of certain genes or regions of the genome will be different between the maternal and paternal alleles, such that certain genes are inactive (imprinted) in one parent and not in the other, and vice versa, based on gender. Any abnormalities in the inheritance of this imprinting pattern (for example, inheriting two copies of the paternal or maternal methylation markers as opposed to one from each) can lead to conditions called methylation disorders. Such disorders cannot be detected by sequencing or deletion/duplication analysis.

A specific methylation assay, such as Methylation-Specific Multiplex Ligation-Dependent Probe Amplification (MS-MLPA), can accurately diagnosis methylation disorders such as Prader–Willi syndrome, Angelman syndrome, Beckwith–Wiedemann syndrome, and Russell–Silver syndrome. MS-MLPA can determine which parental allele is affected. A CMA with a single nucleotide polymorphism (SNP) platform can identify regions of homozygosity (ROH), which may be suggestive of a methylation disorder, as it may signal the inheritance of the same methylation mark from either parent if the ROH is in a region with a known methylation disorder. However, a CMA is unable to determine which parental allele is affected. For most cases where a methylation disorder is suspected, MS-MLPA is typically the most appropriate assay to pursue.

While MS-MLPA is the most appropriate first tier test for methylation disorders, it is not the only test that should be considered, particularly if the MS-MLPA assay is negative. For example, approximately 11% of cases of Angelman syndrome are due to pathogenic sequence variants in the *UBE3A* gene. Knowledge of the molecular basis of the methylation disorder will have a direct impact on genetic counseling and recurrence risks for the family. For example, in the case of Angelman syndrome, if MS-MLPA identifies a maternal deletion in the Angelman syndrome critical region, the recurrence risks for the family is ~1%, while if a pathogenic sequence variant in *UBE3A* is found to be maternally inherited, the recurrence risks will be 50% [16].

### 2.8. Whole Genome Sequencing

The exome represents the protein coding regions of the genome and accounts for approximately 1% of the entire genome. However, approximately 85% of all known pathogenic variants that have been identified so far are located within this protein coding regions of the genome [37]. Whole genome analysis (WGS) sequences the protein coding regions and the remaining ~99% of the genome. While it might initially appear to be beneficial to pursue the larger test from a clinical perspective, there are still limitations associated with this technology. The functional impact of variation within the majority of the noncoding genome outside of the exome has not yet been established and, therefore, the likelihood of identifying a clinically relevant variant is not high, while there can be a significantly large number of VUSs.

It should be noted that many well-established genes have pathogenic variants outside the coding sequences such as intronic variants or variants upstream of the coding regions. Most clinical laboratories make every effort to capture such variants in their exome or panel designs, thus making it less likely for a WGS to have an advantage in this regard. One meta-analysis reported a diagnostic yield ranging from 36–86% for WGS [10]. The diagnostic yield varied between clinical indications with autosomal dominant polycystic kidney disease having a diagnostic yield of 86%. However, it is not clear if the diagnosis was due to only exonic variants, which would have also identified WES; therefore, the diagnostic yield of WGS and WES would have been comparable. If the variants were intronic or in non-coding regions, a difference in the diagnostic yield may occur.

A major advantage of WGS is its ability to capture structural rearrangements, including CNVs, translocations, and inversions using bioinformatics tools, which leverage the ability of WGS to capture those rearrangements’ breakpoints, which are often embedded in the intronic noncoding regions [38]. In addition, several algorithms have been developed to identify repeat expansions and variants in regions of high homology due to pseudogenes (genes which are highly homologous to functional true genes) from WGS data. With its ability to capture SNVs, repeat expansions, CNVs, and other structural rearrangements, WGS promises to become the single genetic test replacing single gene analysis, NGS panels, WES, cytogenetic analysis, and CMA. However, additional work is needed to clinically validate the ability of the new computational tools to robustly capture all the above variants, and to justify its high cost and the data storage issues associated with it. As the cost of WGS decreases in the future and its ability to capture all possible genetic variation improves with time, and as our understanding of non-coding variants increases, it is quite possible that WGS become the comprehensive single-time genetic test [39,40,41].

As previously noted for WES, WGS is a complex test and if possible, that the input of a genetic specialist should be considered, if available.

## 3. Additional Considerations

### 3.1. Interpreting VUS

The presence of a VUS on a report should not automatically be considered a diagnosis for the patient. A VUS should always be considered in the context of the clinical findings. A VUS on a report means the laboratory found a variant in a gene that happens to have a clinical overlap with the reported phenotype. However, if clinical correlation revealed that the gene has a very strong overlap with the patient phenotype, this might prompt the clinician to pursue further investigations to confirm or rule out a role for the VUS in patient disease. For example, a homozygous VUS in *USH2A*, a gene known to cause hearing loss and retinitis pigmentosa, identified in a patient with hearing loss, might prompt the physician to perform an electroretinogram eye exam, which if abnormal, might increase the suspicion that this VUS is clinically significant. It is important to keep in mind that the larger the number of genes that are being analyzed, the higher the possibility of identifying a VUS. There is also an increased chance of identifying a VUS amongst ethnic populations that are underrepresented in control population databases such as gnomAD [42]. If a VUS is identified in a report, the variant should be reviewed annually until it is reclassified.

### 3.2. Negative or Inconclusive Report

It is important to note that our knowledge of the human genes and variants will continue to grow over time and a negative WES or WGS today might turn positive if re-analyzed a year or two later. Similarly, with accumulating evidence over time, a VUS in a panel might be upgraded to pathogenic or likely pathogenic or downgraded to benign or likely benign, leading to significant impacts on patients. It is, therefore, important to check with the testing laboratories about any variant reclassifications especially for patients with highly suspicious VUSs. For WES, when the report is negative, it is generally recommended to request re-analysis within 2–4 years since initial testing to uncover pathogenic variants in novel genes [4,43,44,45].

### 3.3. Diagnostic Yield

Another important consideration when requesting a genetic test is the diagnostic yield for the condition that is being investigated. For conditions that are well established, particularly monogenic conditions, the absence of a pathogenic variant will effectively exclude it from the diagnosis and further evaluations will be required to find a diagnosis. On the other hand, for many conditions that are less genetically characterized, a negative genetic test result does not necessarily exclude the clinical diagnosis. For example, Branchiootorenal (BOR) syndrome has three associated genes, *EYA1*, *SIX1*, and *SIX5*, which account for less than 50% of all clinically confirmed cases [46]. A negative genetic test result is not likely to change the clinical diagnosis for this condition due to it relatively low genetic diagnostic yield. Knowing the diagnostic yield for the condition can determine if additional testing may be required in the event of a negative result or not.

### 3.4. Insurance Coverage

The healthcare service in a given region or country will depend on the level of access the clinician or patient has to genetic testing. For national healthcare systems, patients will often have to meet certain criteria before being eligible for testing. In these situations, it is important to thoroughly evaluate the patient to identify all relevant findings and to determine if the patient is eligible for testing. In private healthcare systems, testing will be dependent on insurance coverage. In some countries, insurance policies may not cover genetic testing or congenital conditions, so it will be important to know this information when pursuing this testing. Many companies will review the insurance claim based on justifications made by the health care provider. Unfortunately, this can be time consuming for providers.

In some healthcare systems, insurance companies may only cover one genetic test, and will not consider stepwise testing, even if it is the most financially optimal testing strategy. In these systems, the considerations for genetic testing discussed above should be combined with considerations about reimbursement models to come up with the most viable testing option at the time. It will still be important to identify the most appropriate genetic test. A simplified way to determine what test to pursue, in this setting, might be to anticipate if any additional follow up testing would be done in the event of a negative result. For example, if the only clinical finding is hearing loss, and a comprehensive hearing loss gene panel is negative, reflexing to a WES analysis will not significantly increase the diagnostic yield, and therefore, a panel would be a suitable first test to order. However, if the panel was negative for another patient with hearing loss and other clinical findings are present, it may be warranted to pursue WES. In this situation, it may be beneficial to start with the most comprehensive test, WES in this case, as it may be subsequently rejected if the hearing loss panel is negative.

## 4. Conclusions

The utilization of genomic testing will only continue to expand in the future across all aspects of medicine due to the continuous novel gene discoveries, the decreasing cost of sequencing, and the development of treatments and cures for previously untreatable genetic conditions. It will be essential for all healthcare providers to have at least a basic understanding of genetic concepts, the various genetic testing strategies available and their advantages and limitations. This knowledge will allow providers to select the most appropriate test for their patient, thereby increasing the likelihood of a diagnosis while reducing the cost and length of time spent on the diagnostic odyssey. However, when possible, the input of a genetic specialist should be sought, especially for complex tests such as WES or WGS.

It is important that providers continue to review genetic results that may have been inconclusive, i.e., if a VUS is reported, the variant should be reviewed at least annually until it is reclassified, and in the case of a negative WES or WGS, a reanalysis of the data should be considered 2–4 years after the initial report.

While this overview is not comprehensive, it will provide providers with the basic knowledge to understand the differences between various genetic testing assays and their limitations.

## Figures and Tables

**Figure 1 genes-12-00818-f001:**
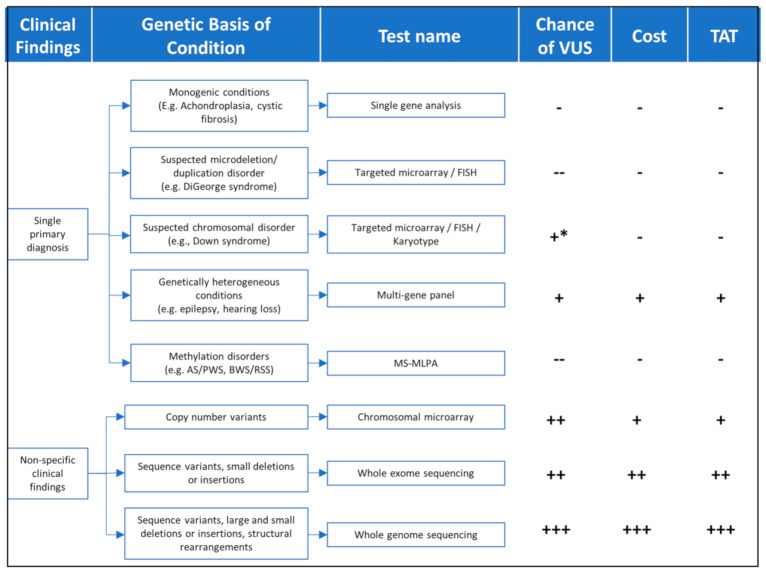
TAT = turnaround time; FISH = Fluorescence in situ hybridization; MS-MLPA = Methylation-Specific Multiplex Ligation-dependent Probe Amplification. Chance of VUS(--) = very low; (-) = Low; (+) = moderate; (++) = high; (+++) = very high; (+ *) moderate risk for karyotype. Cost—(-) = lowest cost (+) = moderate cost (++) = high cost (+++) = highest cost. TAT—(-) = shortest time (+) = moderate time (++) = long time (+++) = longest time.

**Table 1 genes-12-00818-t001:** Overview of common genetic assays.

Test Name	Variants Identified	When to Use	Limitations	Diagnostic Yield	Turnaround Time **	References
Chromosomal microarray	Large chromosomal deletion/duplications High resolution (up to 25 kb)	MCA, ID or DD, and ASD	Cannot identify balanced rearrangements Cannot identify sequence variants or small copy number variants ≤1–10 kb	15–20%	2–6 weeks+	[3]
Karyotype	Large chromosomal deletion/duplications Balanced rearrangements (translocations/inversions) Low resolution (3–5 Mb)	Determine recurrence risks for aneuploidies (e.g., Down syndrome) FHx of Balanced translocations Hx of multiple miscarriages	Low resolution Cannot identify sequence variants or copy number variants <3 MB	~3%	1–2 weeks+	[3]
Single gene sequencing and/or del/dup analysis	Sequence variants, small deletions (<50 bp), or insertions	Phenotype is specific for a single monogenic disorder	Very few conditions are monogenic	Up to 100% depending on clinical ascertainment	1–4 weeks+	
Panels (Preset or customizable)	Sequence variants, small deletions (<50 bp *), or insertions * Larger copy number variants can be identified by some laboratories	Clearly defined phenotype Short differential diagnosis list	Not suitable if no primary phenotype Not suitable if differential diagnosis is long	Variable between condition/phenotype	2–6 weeks+	[4,5,6,7]
Whole exome sequencing	Sequence variants, small deletions (<50 bp *), or insertions (* Larger copy number variants can be identified by some laboratories)	Unclear/large differential diagnosis Non-specific clinical findings Specific test unavailable	Relatively expensive Long turnaround time High chance of VUS Secondary findings Incidental findings	31–53% 8–70% 16–58%	5–12 weeks+	[8,9,10]
Whole genome sequencing	Sequence variants, small deletions, or insertions Structural rearrangements, including balanced translocations, inversions, and deletion/duplications (large and small)	Unclear/large differential diagnosis Non-specific clinical findings Specific test unavailable	Very expensive Long turnaround time High chance of VUS Limited understanding of non-coding variants Secondary findings Incidental findings	8–70% 36–86%	4–16 weeks+	[9,10]
Triplet repeat analysis	Expansion of triplet repeats	Triplet repeat expansions, e.g., Fragile X, myotonic dystrophy, spinocerebellar ataxia	Requires knowledge of conditions caused by triplet repeat expansions	Varies by clinical ascertainment	2–4 weeks+	[11,12,13]
Methylation analysis	Imprinting/methylation disorders/ Uniparental disomy	BWS/RSS, PWS/AS	Requires knowledge of conditions caused by imprinting defects	Variable between conditions	2–4 weeks+	[14,15,16,17]

MCA = multiple congenital anomalies; ID = intellectual disability; DD = developmental delay; ASD = autism spectrum disorder; BWS = Beckwith–Wiedemann syndrome; RSS = Russell–Silver syndrome; PWS = Prader–Willi syndrome; AS = Angelman syndrome; VUS = variant of uncertain significance; FHx = Family history; Hx = History; ** = Turnaround times vary by lab; times stated taken from a survey of labs obtained from concertgenetics.com.

## Data Availability

Not applicable.

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
