# Peer review of "A Simple Practical Guide to Genomic Diagnostics in a Pediatric Setting"

_genes, 2021, doi:10.3390/genes12060818_

Round 1
Reviewer 1 Report
Very good and important review or overview of the current genomic testing field. Very informative.
In the introduction I missed an information about the number of genetic counsellors in the rest of the world (eg Europe, middle East..).
You need to edit the table - explain all abbreviations!!, could be in Leged under the table or in the text.
The editing is needed for Figure 1. also - resolution is bad, abbreviations again, etc.
I think there are to many abbreviations in the text, they are not familiar for "no genetic counsellors".
Author Response
Thank you for your feedback and suggestions on the manuscript. Please see our response below to your suggestions
In the introduction I missed an information about the number of genetic counsellors in the rest of the world (eg Europe, middle East..). - We have included relevant information about the rest of the globe
You need to edit the table - explain all abbreviations!!, could be in Leged under the table or in the text. - table has been updated and now includes all abbreviations
Figure 1 - uploaded with higher resolution
I think there are to many abbreviations in the text, they are not familiar for "no genetic counsellors". - We believe that we need to include all abbreviations, as these abbreviations are commonly used within genetics. We believe it will be important for providers to become familiar with these concepts as they pursue genetic testing
We appreciate your time for reviewing our manuscript and thank you for feedback. Your suggestions along with the other reviewers edits have been incorporated.
Reviewer 2 Report
Please find the attached review.

Author Response
Thank you for taking the time to review our manuscript and providing feedback and suggestions. Please see our response to your suggestions below
I would recommend to make the paragraphs more concise and to add the clinical examples in all paragraphs. The authors did it in most of the explanations of different genetic diagnostic tests, which is very useful for clinicians. - Paragraphs edited to reduce complexity. Additional clinical examples have been included where it was deemed appropriate
I think the remark about the necessity of written informed consent for patient/parent for all the diagnostic tests should be more clearly written. - This section was edited and expanded for clarity
In my opinion some of the more complicated tests still need the opinion of a geneticist and it may be impossible to interpret the results without this expertise. - This note was included and has been mentioned in the final conclusion
Table - new higher resolution table has been uploaded, additional information has been included to clarify the table at the bottom. No reference was inlcuded for the single gene analysis. Due to the number of genes that can be analyzed individuals, no reference would be suitable. references for whole exome and whole genome diagnostic yeild have been separated for clarity on respective references.
Figure 1 - New figure uploaded with higher resolution. Methylation test inlcuded.
Statement written in lines 242-250 is very important and could be emphasised in Conclusions. - statement has been included in the conclusion
Lines 250-252- unclear text. - moved to new section "3.3" for clarity
Line 293- MS-MLPA test for diagnosis of Prader-Willi syndrome is only the first step. This type of diagnostic tool does not allow to set the precise diagnosis, which is important not only for genetic counselling but also for the clinical management of patients - additional information provided for clarity
Your time and feedback is much appreciated. Your edits and the suggestions from other reviewers have been incorporated.
Reviewer 3 Report
Molecular genetic/genomic testing has become increasingly prevalent in healthcare systems around the world. It is a powerful diagnostic tool that guides important decision-making processes for providers. Therefore, having a high-level yet competent knowledge base of various genomic tests is important for healthcare providers to accelerate diagnosis, reduce errors and improve patient care.
This review by Taylor and colleagues offers an overview of test strategies and their suitable clinical scenarios, provides a framework for how each test should be ordered, and touches upon evolving issues such as variants of unknown significance and insurance coverage for genomic testing.
This manuscript, which is clearly written and strikes a balance between technical details and practical applications, should appeal to its intended readership—primary pediatric healthcare providers. It should also be an informative read for the audience of Genes.
I think this paper will be further strengthened by addressing these following points.
Major:
- Table 1: Single-gene sequencing: could the authors comment on why small deletions have to be <50bp? Sanger sequencing should be able to detect small deletions that are larger than 50bp. Also, the authors need to provide a lower bound for yield.
- Table 1: Whole-exome sequencing: could the authors provide references for the three different yield ranges?
- Table 1: The authors may want to attach a dollar amount to “relatively expensive” and “very expensive.”
- Figure 1: please use an image with higher resolution. It is blurry at 125% or 150% zoom.
- The authors may want to provide a box of glossary, instead of explaining technical terms in the main text (an example: lines 127 – 128).
- Lines 141-144: the authors may want to cite a couple of representative papers/reviews on calling structural variations from NGS data. In addition, please also cite the source that suggests WES should be considered as the first-tier test for patients with neurodevelopmental disorders.
- Line 180: the authors should cite the American College of Medical Genetics and Genomics recommendation for the 59 genes included in the secondary findings.
- Line 347: the authors might want to cite the following papers here or somewhere else:
https://www.cell.com/fulltext/S0092-8674(12)00166-3
https://www.ncbi.nlm.nih.gov/pmc/articles/PMC1681452/
- In the insurance section, the authors are encouraged to discuss that there might very well be a public health/preventive medicine motivation for governments to subsidize or sponsor population-scale whole genome sequencing. Perhaps cite the influential initiatives in Iceland and in the UK:
https://www.nature.com/articles/ng.3247
https://www.genomicsengland.co.uk/about-genomics-england/the-100000-genomes-project/ (perhaps cite one of the papers from this initiative)
- The authors are strongly encouraged to cite the following pertinent papers:
https://www.frontiersin.org/articles/10.3389/fped.2020.00373/full
https://www.nejm.org/doi/10.1056/NEJMra1711801
https://www.nejm.org/doi/10.1056/NEJMra1312543
- The authors might want to consider briefly introducing technologies on the horizon (i.e. nanopore seq) and how such technique can help fill the current gap in terms of test capabilities at the end of this article.
Minor:
- Lines 39 -41: the authors may want to discuss the relationship between the shortage of genetic counsellors and the extent of adoption of clinical genetic testing in healthcare systems outside of North America. Is there a high adaption rate of clinical genetic testing but a shortage of genetic counsellors? Or is there only limited adaption of clinical genetic testing (along with the lack of skilled technicians and clinical scientists), rendering a limited demand for genetic counsellors?
- Lines 291-292: the authors are encouraged to summarize the exact molecular test for triplet repeat disorders.
- Lines 331-332: the authors might want to briefly discuss why this is the case. What would have caused a disease variant in exonic region to escape detection by WES but not WGS?
- Lines 336-341: Is the segment also referring to ref 29?
- Lines 250-252: These two sentences are a bit out of the place.
- Lines 57-61: this is one of the examples where the authors might want to break a very long sentence into shorter ones to improve readability.
- Line 20: “their patient” perhaps “their patients”
- Line 46: perhaps “one-by-one” and “gene-by-gene”
- Lines 70-71: maybe rephrase it as the “scale of testing (i.e. targeted panel vs. exome/whole genome)”
- Lines 171-173: needs clarifications.
- Line 239: refs 16 and 17 need to be in “[ ]”
- Line 243: “within” should be “will”
- Line 268: “it is might be” should be “it might be”
- Line 303: perhaps “cannot be detected by conventional sequencing…”
Author Response
Thanks you for your detailed review and suggestions on our submission. In response to your suggestions - please see below
- Table 1: Single-gene sequencing: could the authors comment on why small deletions have to be <50bp? Sanger sequencing should be able to detect small deletions that are larger than 50bp. Also, the authors need to provide a lower bound for yield. -
To our knowledge it generally accepted that resolution of sequencing is around 50bp for sequencing technologies. Anything above can be done differently (MLPA, PCR/gel, ddPCR, or even CNVs for NGS coverage). several labs won’t recommend using Sanger for deletions above 50bp which is why we have listed this number as the limitation
- Table 1: Whole-exome sequencing: could the authors provide references for the three different yield ranges? - References have been separated to to correspnd to each yield range
- Table 1: The authors may want to attach a dollar amount to “relatively expensive” and “very expensive.” - this was something that we could not easily could do and would require its own separate analysis. The issue being we are aiming this for a global audience, and many providers will not have access to laboratories outside of their home country. In addition, there are large genetic testing companies in America that are providing testing at a much reduced cost compared to the market average, making these outliers difficult to include in the analysis. Instead, we want providers to understand the cost associated with each testing compared to other types of genetic testing. A direct comparison in the cost would be easier for sequencing assays
- please use an image with higher resolution. It is blurry at 125% or 150% zoom. - new image with higher resolution uploaded
- The authors may want to provide a box of glossary, instead of explaining technical terms in the main text (an example: lines 127 – 128) - we believed that as trisomy 21 is one of the most common chromosomal conditions, it would be worthwhile to review the possible inheritance patterns assoicated with this condition.
- Lines 141-144: the authors may want to cite a couple of representative papers/reviews on calling structural variations from NGS data. In addition, please also cite the source that suggests WES should be considered as the first-tier test for patients with neurodevelopmental disorders. - citations included and clarified
- Line 180: the authors should cite the American College of Medical Genetics and Genomics recommendation for the 59 genes included in the secondary findings. - citations included
- Line 347: the authors might want to cite the following papers here or somewhere else: - citations included
- In the insurance section, the authors are encouraged to discuss that there might very well be a public health/preventive medicine motivation for governments to subsidize or sponsor population-scale whole genome sequencing. Perhaps cite the influential initiatives in Iceland and in the UK: - while we concur about the importance of governments recognizing the importance of genetic testing as a public health initiative, it is outside the purview of this review.
- The authors are strongly encouraged to cite the following pertinent papers: - unfortunately, I was unable to access the NEJM papers to review and include
- The authors might want to consider briefly introducing technologies on the horizon (i.e. nanopore seq) and how such technique can help fill the current gap in terms of test capabilities at the end of this article. - The aim of this article was to provide an overview of current technologies and help support providers in test selection. As such, we believe that reviewing future technologies was outside the scope of this review.
- Lines 39 -41: the authors may want to discuss the relationship between the shortage of genetic counsellors and the extent of adoption of clinical genetic testing in healthcare systems outside of North America. Is there a high adaption rate of clinical genetic testing but a shortage of genetic counsellors? Or is there only limited adaption of clinical genetic testing (along with the lack of skilled technicians and clinical scientists), rendering a limited demand for genetic counsellors? - This is a very insightful question that we are unable to answer. We do not know if there is a lack of uptake of genetic testing in regions with a low per capita of genetic specialists. such a question would require a deeper investigation. We want to highlight that there is a relative shortage of genetic specialists and will continue to be for sometime.
- Lines 291-292: the authors are encouraged to summarize the exact molecular test for triplet repeat disorders. - The name of the testing we believed could cause confusion and not help providers understand that specialist testing is needed for triplet repeat disorders and these disorders can not be detected by other technologies
- Lines 331-332: the authors might want to briefly discuss why this is the case. What would have caused a disease variant in exonic region to escape detection by WES but not WGS? - section re-written to provide clarification
- Lines 250-252: These two sentences are a bit out of the place. - information moved to its own section - "3.2 Negative or inconclusive report"
- Lines 57-61: this is one of the examples where the authors might want to break a very long sentence into shorter ones to improve readability. - rewritten for clarity
- Line 20: “their patient” perhaps “their patients” - fixed
- Line 46: perhaps “one-by-one” and “gene-by-gene” - fixed
- Lines 70-71: maybe rephrase it as the “scale of testing (i.e. targeted panel vs. exome/whole genome)” - reworded
- Lines 171-173: needs clarifications. - rewritten for clarificaiton
- Line 239: refs 16 and 17 need to be in “[ ]” - fixed
- Line 243: “within” should be “will” - fixed
- Line 268: “it is might be” should be “it might be” - fixed
- Line 303: perhaps “cannot be detected by conventional sequencing…” - we kept the original statement as we felt that sequencing would include both NGS and sanger sequencing
Many thanks for your input and suggestions. You edits have been included along with the other reviewers suggestions.